# Using genome and transcriptome data from African-ancestry female participants to identify putative breast cancer susceptibility genes

African-ancestry (AA) participants are underrepresented in genetics research. Here, we conducted a transcriptome-wide association study (TWAS) in AA female participants to identify putative breast cancer susceptibility genes. We built genetic models to predict levels of gene expression, exon junction, and 3′ UTR alternative polyadenylation using genomic and transcriptomic data generated in normal breast tissues from 150 AA participants and then used these models to perform association analyses using genomic data from 18,034 cases and 22,104 controls. At Bonferroni-corrected $P < 0.05$, we identified six genes associated with breast cancer risk, including four genes not previously reported (CTD-3080P12.3, EN1, LINC01956 and NUP210L). Most of these genes showed a stronger association with risk of estrogen-receptor (ER) negative or triple-negative than ER-positive breast cancer. We also replicated the associations with 29 genes reported in previous TWAS at $P < 0.05$ (one-sided), providing further support for an association of these genes with breast cancer risk. Our study sheds new light on the genetic basis of breast cancer and highlights the value of conducting research in AA populations.

Breast cancer is the most common cancer diagnosed among women in the world, with ~2.3 million new cases diagnosed annually, accounting for ~12% of all new cancer diagnoses[1]. In the United States, women with African-ancestry (AA) generally have a slightly lower incidence rate of breast cancer than women with European-ancestry (EA)[2]. However, AA women are at a higher risk of developing estrogen receptor (ER) negative and triple-negative breast cancer (TNBC) than their EA counterparts[3,4]. It has been suggested that genetic factors may explain some of the difference in breast cancer risk between these two populations[5–8].

Since 2007, genome-wide association studies (GWAS) have identified >200 genetic variants associated with breast cancer risk[9–12]. However, most of these genetic variants were identified from studies conducted in populations of European or Asian ancestry[13,14]. Recently, several transcriptome-wide association studies (TWAS) have been conducted to identify putative susceptibility genes for breast cancer[9,15–18]. However, none of these studies were conducted on AA participants. Two recent TWAS included AA female participants. However, these two studies were small, evaluated only breast cancer mortality and recurrence, and used transcriptomic data from tumor tissues for model building, which could affect the accuracy of model building[19,20]. Given the differences in genetic architecture between AAs and other populations[21], genetic studies conducted among AA participants could provide added insights into breast cancer genetics and biology.

Previous TWAS of breast cancer focused primarily on investigating associations of gene expression levels with the risk of this common cancer. It has been proposed that investigating genetic variants related to post-transcriptional regulations, such as performing splicing TWAS (spTWAS) to investigate exon junction levels or performing alternative

✉e-mail: wei.zheng@vanderbilt.edu

polyadenylation (APA)-wide association studies (APA-WAS) to investigate 3′ untranslated region (3′ UTR) APA levels, could also help to identify putative disease susceptibility genes[22,23]. In this study, we performed these three types of TWAS in AA participants to systematically investigate genes associated with breast cancer risk.

## Results

We used GWAS data from the African-ancestry Breast Cancer Genetic (AABCG) Consortium. Study participants were recruited from more than 20 studies conducted in the United States and Africa, the details of which were reported in a previous publication and summarized in the supplemental notes and Table S1[24]. In brief, 18,034 female cases and 22,104 female controls of African-ancestry were included in this study (Table S2). Information regarding breast cancer subtypes was available for 9304 ER-positive cases, 4924 ER-negative cases, and 2860 TNBC cases.

We performed RNA sequencing of normal breast tissue samples donated by 150 AA female participants to the Susan G. Komen Normal Tissue Bank. Genomic DNA samples of these participants were genotyped using the Illumina multi-ethnic genotyping array (MEGA) platform. These data were used to build genetic models to predict levels of gene expression, APA, and exon junction.

### Association analyses identify new putative susceptibility genes for breast cancer risk

Of the 36,387 genes measured, gene expression prediction models were successfully built for 18,787 genes, of which 9982 genes achieved a five-fold cross-validation prediction performance with $R > 0.1$ (Table 1). Applying these 9982 models to GWAS data using S-PrediXcan, one lncRNA gene *CTD-3080P12.3* (also known as *TERLR1*) was associated with overall breast cancer risk at the Bonferroni-corrected significance of $P < 5.0 \times 10^{-6}$ (0.05/9982). We further evaluated the associations between predicted gene expression and risk of breast cancer by subtypes and identified two genes, EN1 and LINC01956 (for ER-negative and TNBC), showing an association at the Bonferroni-corrected significance level (Table 2). These three genes (*CTD-3080P12.3, EN1,* and *LINC01956*) have not been previously identified in TWAS for breast cancer risk. *CTD-3080P12.3* is located at a known breast cancer risk locus (lead variant rs2853669). Both *EN1* and *LINC01956* are located at an AA-specific risk locus (lead variant rs76664032) that was identified in our recent GWAS conducted among AA participants[24].

We identified 22,559 APA events and 80,566 exon junctions, of which 3309 APA events and 11,426 exon junctions can be predicted with an $R > 0.1$ (Table 1). Using these models, we identified one APA event in gene *TET2* that showed an association with overall breast cancer risk at the Bonferroni-corrected significance of $P < 1.5 \times 10^{-5}$ (0.05/3309, Table 2). We also found that two exon junctions in genes *BRD9* and *NUP210L* showed an association with overall breast cancer risk at the Bonferroni-corrected significance of $P < 4.3 \times 10^{-6}$ (0.05/11,426, Table 2). Both *TET2* and *BRD9* are located at previously GWAS-identified breast cancer risk loci (lead variants rs62331150 and rs2853669). *NUP210L* was not located at any known breast cancer risk locus and has not been reported by any previous breast cancer TWAS. Analyses by subtypes of breast cancer identified *TET2* from APA-WAS in association with ER-negative breast cancer risk at the Bonferroni-corrected significant level and *BRD9* from spTWAS in association with ER-negative and TNBC subtypes at the Bonferroni-corrected significant level (Table 2).

We further assessed genetic variants located within the ±500Kb region for each of the 13 genes that were identified in our TWAS to be significantly associated with breast cancer risk, overall or by subtypes. Within these regions, 11 lead variants show GWAS significance at $P < 5 \times 10^{-8}$ (Table 3) in AA with different subtypes of breast cancer. To evaluate whether the associations for these 13 genes might be explained by these lead risk variants identified in GWAS, we performed conditional analyses by adjusting for the nearest GWAS-identified risk variants. The *NUP210L* gene is located far away from any of the index risk variants, and thus we cannot perform a conditional analysis for this gene. For gene *EN1* and *LINC01956*, we adjusted the SNP rs76664032 identified in our recent AA GWAS. SNP rs76664032 is located at 2q14.2 and is 10Kb from the 3′ end of *LINC01956* and 18Kb from the 3′ end of *EN1*. The results of the conditional analysis revealed that the association with EN1 and LINC01956 remains highly significant for the risk of ER-negative (EN1: $P_{\text{Conditional}} = 5.0 \times 10^{-7}$; LINC01956: $P_{\text{Conditional}} = 6.4 \times 10^{-4}$) and TNBC subtypes (EN1: $P_{\text{Conditional}} = 4.0 \times 10^{-7}$; LINC01956: $P_{\text{Conditional}} = 7.1 \times 10^{-4}$) (Table S3). These results suggest that the association with *EN1* and *LINC01956* identified in our TWAS cannot be explained entirely by the risk variant identified in GWAS. On the other hand, the association was substantially attenuated for the *CTD-3080P12.3* gene ($P_{\text{Conditional}} = 0.028$) or became non-significant for other genes, suggesting that most of the association signal for these genes can be explained by the index variants identified in GWAS. We also performed permutation tests to evaluate the extent to which the risk variant-expression weights contribute to the TWAS association signals identified in this study and demonstrated that the integration of expression data significantly enhances the association with breast cancer risk (Table S3). We utilized eQTpLot[25] to visualize eQTLs and GWAS statistics for putative risk genes identified in this study. All the variants employed in the prediction models were used in the visualization (Supplementary Figs. S1–S11). For many of these genes, variants used in the prediction models showed a highly significant association with breast cancer risk.

### Evaluation of previously TWAS-identified genes

We used AABCG data to evaluate whether genes identified in our previous TWAS conducted among Asian- and European-ancestry participants[9] are also associated with breast cancer risk in AA participants. Of the 137 genes identified in our previous TWAS (Table S4), 106 genes (77%) can be predicted in our genetic models for levels of gene expression, APA event, and exon junction, and 86 genes (63%) reach prediction $R > 0.1$ (Table S4). Of them, 29 genes showed the same association direction as the previous Asian- or European-ancestry TWAS at $P < 0.05$ (one-sided) (Table 4). We also evaluated genetic variants within the ±500 Kb region for each of these 29 replicated genes. Only one lead variant (rs56069439 of *ABHD8*) in AA participants showed an association with breast cancer risk at $P < 5 \times 10^{-8}$ (Table S5).

## Discussion

Here, we have conducted a TWAS in African-ancestry participants to systematically evaluate the associations of genetically predicted transcriptome levels with breast cancer risk using expression data from normal breast tissues. We identified six genes associated with the risk of breast cancer, overall or by subtypes, at the Bonferroni-corrected significant level of 0.05, including four genes not yet reported previously. Perhaps more importantly, most of these genes showed a more specific association with ER-negative or TNBC subtypes, for which AA participants are at a higher risk than participants of other racial groups. In addition, we replicated the association for 29 genes identified from the previous Asian- or European-ancestry TWAS, providing support for a potential causal association of these genes with breast cancer risk. Our study highlights the value of conducting genetic studies in the AA population and provides significant new information to understand the genetics and etiology of breast cancer in AA participants who are significantly underrepresented in genetic studies.

We identified four new genes (*CTD-3080P12.3, EN1, LINC01956,* and *NUP210L*) showing a significant association with breast cancer risk at the Bonferroni-corrected significance level. *CTD-3080P12.3* was associated with overall breast cancer risk. *CTD-3080P12.3*, also known

**Table 1 | Summary of genetic models built to predict levels of gene expressions, alternative polyadenylation (APA) event, and exon junction**

| Models | No. of events for model building | No. of predictable events | No. of events predicted with R > 0.1 |
|---|---|---|---|
| Gene expression | 36,387 | 18,787 | 9982 |
| APA event | 22,559 | 8120 | 3309 |
| Exon junction | 80,566 | 31,591 | 11,426 |

**Table 2 | Genes associated with breast cancer risk at the Bonferroni-corrected significant level of 0.05**

| Breast cancer subtype[a] | Gene | Cytoband | Nearest index SNP[b] | Z score | P value |
|---|---|---|---|---|---|
| Exp-TWAS | | | | | |
| Overall | CTD-3080P12.3 | 5p15.33 | rs2853669 | 4.7 | $3.2 \times 10^{-6}$ |
| ER-Neg[d] | EN1 | 2q14.2 | rs76664032[c] | −5.7 | $1.6 \times 10^{-8}$ |
| | LINC01956 | 2q14.2 | rs76664032[c] | −5.4 | $5.3 \times 10^{-8}$ |
| TNBC[d] | EN1 | 2q14.2 | rs76664032[c] | −5.8 | $6.9 \times 10^{-9}$ |
| | LINC01956 | 2q14.2 | rs76664032[c] | −5.6 | $2.5 \times 10^{-8}$ |
| APA-WAS | | | | | |
| Overall | TET2 | 4q24 | rs62331150 | −4.4 | $1.2 \times 10^{-5}$ |
| ER-Neg | TET2 | 4q24 | rs62331150 | −4.9 | $1.2 \times 10^{-6}$ |
| spTWAS | | | | | |
| Overall | BRD9 | 5p15.33 | rs2853669 | −4.7 | $2.3 \times 10^{-6}$ |
| | NUP210L | 1q21.3 | / | −4.7 | $3.0 \times 10^{-6}$ |
| ER-Neg | BRD9 | 5p15.33 | rs2853669 | −5.2 | $2.2 \times 10^{-7}$ |
| TNBC | BRD9 | 5p15.33 | rs2853669 | −4.8 | $1.8 \times 10^{-6}$ |

*P* values were derived from the Z score tests (two-sided). Statistical significant threshold for multiple comparison adjustment is defined as $P < 5.0 \times 10^{-6}$ for gene expression TWAS of 9982 tests (0.05/9982), $P < 1.5 \times 10^{-5}$ for APA-WAS of 3309 tests (0.05/3309), and $P < 4.3 \times 10^{-6}$ for spTWAS of 11,426 tests (0.05/11,426) using Bonferroni correction.

[a]*ER-Pos* estrogen receptor (ER)-positive, *ER-Neg* ER-negative, *TNBC* triple-negative breast cancer, *TWAS* transcriptome-wide association study, *Exp-TWAS* gene expression TWAS, *APA-WAS* alternative polyadenylation (APA)-wide association study, *spTWAS* splicing TWAS.

[b]Within ± 500 Kb region.

[c]This index SNP (rs76664032) was identified in our recent GWAS conducted among AA female participants.

[d]Suggestive associations were found for genes CTD-3080P12.3 ($P = 1.3 \times 10^{-5}$) with ER-negative breast cancer risk and MRPL34 ($P = 9.2 \times 10^{-6}$) with TNBC risk at the p value near the significant threshold after Bonferroni correction.

as *TERLR1*, is located at a well-known breast cancer risk locus (lead variant rs2853669, also known as the *TERT* SNP), however, this gene has not been reported in association with breast cancer in any previous studies. The upregulation of this lncRNA was reported to enhance carcinogenesis and metastasis of esophageal squamous cell carcinoma[26]. *EN1* and *LINC01956* were associated with the risk of ER-negative breast cancer and TNBC, respectively. These two genes are located at an AA-specific risk locus identified by our recent GWAS conducted among AA participants[24]. *EN1* is a transcription factor with known roles in brain[27] and dermomyotome[28] development and its downregulation significantly reduced viability and tumorigenicity in TNBC cell lines[29]. LincRNA *LINC01956* is an *E2F1* target gene, and its overexpression correlates with poor prognosis in basal-like breast cancer participants[30]. *NUP210L* was found to be associated with overall breast cancer risk. It has been reported that *NUP210L* could have a significant impact on the function of nuclear pores, which may play a role in defining the behavior of breast cancer[31]. Further in vitro functional experiments may be applied to evaluate the potential functions of these genes.

In this study, we replicated the associations for 29 putative breast cancer risk genes identified in our previous TWAS conducted in European or Asian participants at nominal $P < 0.05$, providing additional evidence to support the associations of these genes with breast cancer risk. Although the present study used the largest GWAS conducted among AA participants, to our knowledge, the sample size is still relatively small compared with GWAS studies conducted among European or Asian participants. The more limited sample size may be a potential reason for the failure to replicate some previously reported putative breast cancer susceptibility genes. Although these results do not prove causality, consistent findings from studies conducted in different populations provide support for a potential true association, particularly given that the lead risk variants in these populations are different.

In this study, we built three types of prediction models: gene expression, APA events, and exon junctions. Of note, three genes showing a Bonferroni-corrected significant association were identified in APA-WAS or spTWAS, including one new gene (*NUP210L*) not reported before. As APA-WAS or spTWAS investigate genetic variants related to post-transcriptional regulations, these genetic variants may affect the breast cancer risk of AA through post-transcriptional and regulatory mechanisms. The conventional TWAS method, which focuses only on gene expression, may fail to identify transcriptional events related to post-transcriptional regulation. Studies using APA-WAS and spTWAS, along with traditional TWAS, can potentially identify additional genes associated with complex traits.

The major purpose of TWAS is to identify breast cancer susceptibility genes, similar to breast cancer GWAS that aim at identifying genetic variants for breast cancer risk. Previous studies used gene expression data to build gene prediction models for TWAS[9,15,32]. Gene expressions in tumor tissue are altered by somatic mutations, and thus, their levels are not less likely to be regulated by germline variants, which could introduce noise and affect the accuracy of model building. We used gene expression data from normal breast tissues to build gene expression prediction models for TWAS, which is a major strength of this study. However, although TWAS identifies genes that are associated with the trait of interest, they do not prove that these genes are causal. Future studies, including functional experiments, are needed to generate additional evidence for causal inference.

In summary, we performed a comprehensive TWAS based on the integration of large GWAS data and transcriptomic data from African-ancestry participants, to identify candidate susceptibility genes associated with breast cancer risk. Our findings improve the population diversity in genetic studies for breast cancer and provide new insights into the biology and genetics of breast cancer.

## Methods

Normal breast tissue samples donated by 150 African-ancestry female participants to the Susan G. Komen Normal Tissue Bank were used in this study to generate genomic and transcriptomic data to build genetic models to predict levels of gene expression, APA, and exon junction. Normal breast biopsies collected from the upper outer quadrant of the breasts were frozen within an average of six minutes from the time of biopsy, allowing for the preservation of the tissue quality. All donors provided written informed consent, and the protocol was approved by the Indiana University Institutional Review Board.

Genomic DNA samples of these 150 AA participants were genotyped using the Illumina MEGA platform at Vanderbilt University Medical Center. All individuals had at least an 80% proportion of African ancestry using Admixture[33], with 1000 Genome samples as reference (Table S6). For model building, we excluded genetic variants with minor allele frequency <5%, Hardy–Weinberg equilibrium *p* value $< 10^{-4}$ and missing genotyping rate >5%. SNPs with a consistency rate <98% among duplicate samples were also excluded. All

**Table 3 | Association results for the lead variants within ±500 Kb regions of the 13 putative breast cancer risk genes identified in this study, results from the African-ancestry Breast Cancer Genetic Study**

| Breast cancer subtype | Gene | Lead variant | Cytoband | Position (hg38) | Allele[a] | EAF[b] | OR (95% CI) | P value |
|---|---|---|---|---|---|---|---|---|
| Overall | CTD-3080P12.3 | rs10069690 | 5p15.33 | 1279675 | T/C | 0.59 | 1.14 (1.11, 1.18) | $1.07 \times 10^{-16}$ |
| | TET2 | rs61751053 | 4q24 | 105613442 | T/C | 0.01 | 1.49 (1.30, 1.70) | $1.07 \times 10^{-8}$ |
| | BRD9 | rs10069690 | 5p15.33 | 1279675 | T/C | 0.59 | 1.14 (1.11, 1.18) | $1.07 \times 10^{-16}$ |
| | NUP210L | rs1411273 | 1q21.3 | 153629959 | T/G | 0.71 | 1.08 (1.05, 1.12) | $5.00 \times 10^{-6}$ |
| ER-negative | EN1 | rs76664032 | 2q14.2 | 118823485 | A/G | 0.81 | 1.22 (1.15, 1.30) | $1.37 \times 10^{-9}$ |
| | LINC01956 | rs76664032 | 2q14.2 | 118823485 | A/G | 0.81 | 1.22 (1.15, 1.30) | $1.37 \times 10^{-9}$ |
| | CTD-3080P12.3 | rs10069690 | 5p15.33 | 1279675 | T/C | 0.59 | 1.30 (1.23, 1.37) | $1.71 \times 10^{-24}$ |
| | BRD9 | rs10069690 | 5p15.33 | 1279675 | T/C | 0.59 | 1.30 (1.23, 1.37) | $1.71 \times 10^{-24}$ |
| | TET2 | rs57478337 | 4q24 | 105645241 | A/G | 0.02 | 1.39 (1.16, 1.67) | $3.38 \times 10^{-4}$ |
| TNBC | EN1 | rs76664032 | 2q14.2 | 118823485 | A/G | 0.80 | 1.30 (1.20, 1.42) | $3.51 \times 10^{-10}$ |
| | LINC01956 | rs76664032 | 2q14.2 | 118823485 | A/G | 0.80 | 1.30 (1.20, 1.42) | $3.51 \times 10^{-10}$ |
| | MRPL34 | rs12974508 | 19p13.1 | 17290712 | T/C | 0.41 | 0.73 (0.68, 0.77) | $1.29 \times 10^{-23}$ |
| | BRD9 | rs10069690 | 5p15.33 | 1279675 | T/C | 0.59 | 1.38 (1.30, 1.48) | $7.31 \times 10^{-24}$ |

[a]Effect allele/other allele.
[b]Effect allele frequency among controls.

genotyping data were imputed using the Trans-Omics for Precision Medicine (TOPMed) as a reference panel. Genetic variants with imputation quality score ($r^2$) < 0.8 were excluded. Since it is difficult to determine the effect allele, multi-allelic SNPs and strand-ambiguous SNPs (with alleles A/T or C/G) were also excluded. Finally, SNPs that had not been included in the final analysis data set of the AABCG GWAS were excluded for model building.

Total RNA was extracted and purified using Qiagen's AllPrep DNA/RNA/miRNA Universal Kit (Qiagen), following the manufacturer's instructions. The quantity and quality of the DNA/RNA samples were checked by Nanodrop (E260/E280 and E260/E230 ratio) and by separation on an Agilent BioAnalyzer. Rnase H was used to remove rRNA. Each sample was sequenced pair-ended with a read length of 100 bp using DNBSEQ on BGISeq. A minimum of 10 M reads were obtained for each sample.

Summary-level statistics data from the African-ancestry Breast Cancer Genetic Study (AABCG), a GWAS conducted on female participants of African ancestry recruited from more than 20 studies in the U.S. and Africa, were used for the association analysis. Detailed descriptions of participating studies are included in the Supplementary Materials. All study participants provided informed consent, and the AABCG was approved by the Vanderbilt University Medical Center institutional review board. All participating studies were approved by their appropriate ethics or institutional review board.

## Transcriptome data profiling and processing

**Gene expression.** RNA sequencing (RNA-Seq) data were processed following the mRNA analysis pipeline of the genotype-tissue expression (GTEx) project[34]. A two-pass method of the Spliced Transcripts Alignment to a Reference (STAR) software was used for raw data alignment to the human reference genome (hg38)[35]. The GENCODE version 26 was used for coding gene and noncoding RNAs annotation in the human genome[36]. Gene expression levels were quantified from aligned BAM files using RNAseQC v1.1.9[37]. A gene was removed if it did not express in more than 5% of all samples. Gene expression levels were measured using transcript per million (TPM) and log2-transformed. Quantile normalization was performed to standardize the expression level across 150 samples to the same scale. Probabilistic estimation of expression residuals (PEER) factors were calculated to correct for batch effects and other potential experimental confounders in further model building[38,39]. Thirty PEER factors were

determined as a function of sample size as suggested in the GTEx protocol previously[34].

**APA events.** We quantified levels of APA events by using the Percentage of Distal polyA site Usage Index (PDUI) estimated from DaPars v2.0 for each sample from aligned RNA-Seq data[40]. An APA event was removed if it had >5% missing values among all samples. We performed quantile normalization to transform the quantified PDUI values of APA for each sample to the same distribution. PEER factors were estimated by using the normalized APA levels to correct batch effects and experimental confounders in our downstream prediction model building. Thirty PEER factors were determined as suggested.

**Exon junctions.** We estimated exon junction levels from aligned RNA-Seq data by using a probabilistic framework, MISO (Mixture-of-Isoforms)[41]. Percent Spliced Isoform (PSI, $\Psi$ value) was used to quantify the isoform expression. An exon junction was removed if it had >5% missing values among all subjects. Quantile normalization was performed to the PSI values, and thirty PEER factors were estimated to correct for batch effects and experimental confounders in our further prediction model building.

## Building genetic prediction models

Genetic prediction models for each type of transcriptomic event (gene expression, APA event, and exon junction) were built using the processed genotype and quantified levels (TPM, PDUI, and PSI, respectively). Age, PEER factors, and the top five genetic principal components were adjusted in the model building. All flanking genetic variants (±500 Kb region of the respective transcriptomic event) available in the GWAS data of breast cancer were used to build the elastic net model implemented in the *glmnet* R package, with $\alpha = 0.5$, as recommended by Gamazon et al.[42]. Five-fold cross-validation was used to validate the models internally. Prediction R values (the correlation between predicted and observed quantified level) were used to estimate the prediction performance of each prediction model.

## Association analyses with breast cancer risk using GWAS data

Based on the weight matrix and the summary statistics data on SNPs from GWAS data, we evaluated the association between genetically predicted transcriptome levels (gene expression, APA levels, or exon junction levels) and breast cancer risk using the method from the

**Table 4 | Replications of previous TWAS-identified breast cancer-associated genes in African-ancestry participants at a nominal significant level of $P < 0.05$ with consistent direction**

| Gene | Current study | | | | Previous TWAS (EUR/ASN) | | |
|---|---|---|---|---|---|---|---|
| | Z score | P value | Breast cancer subtype[a] | Model[b] | Z Score | P Value | Breast cancer subtype[a] |
| ABHD8 | 2.4 | $9.0 \times 10^{-3}$ | Overall | Exp-TWAS | 4.8 | $2.1 \times 10^{-6}$ | Overall |
| ALS2CR12 | 1.9 | $2.9 \times 10^{-2}$ | Overall | spTWAS | 8.4 | $5.5 \times 10^{-17}$ | Overall |
| ATG10 | −1.9 | $2.7 \times 10^{-2}$ | ER-Pos | APA-WAS | −6.7 | $2.5 \times 10^{-11}$ | Overall |
| CASP8 | −2.4 | $7.9 \times 10^{-3}$ | Overall | Exp-TWAS | −8.5 | $2.0 \times 10^{-17}$ | Overall |
| CMB9-22P13.1 | 2.2 | $1.4 \times 10^{-2}$ | ER-Neg | Exp-TWAS | 5.6 | $1.7 \times 10^{-8}$ | Overall |
| CPNE1 | −2.4 | $9.2 \times 10^{-3}$ | Overall | Exp-TWAS | −4.7 | $2.9 \times 10^{-6}$ | Overall |
| GSTM1 | −2.0 | $2.5 \times 10^{-2}$ | Overall | spTWAS | −4.8 | $1.4 \times 10^{-6}$ | Overall |
| GTF2IP1 | −1.8 | $3.4 \times 10^{-2}$ | Overall | Exp-TWAS | −4.8 | $1.3 \times 10^{-6}$ | Overall |
| GTF2IRD2 | 1.9 | $3.1 \times 10^{-2}$ | Overall | Exp-TWAS | 5.2 | $1.8 \times 10^{-7}$ | Overall |
| HLA-F | 1.7 | $4.2 \times 10^{-2}$ | Overall | APA-WAS | 4.8 | $2.1 \times 10^{-6}$ | Overall |
| KLHDC7A | −1.7 | $4.7 \times 10^{-2}$ | Overall | Exp-TWAS | −6.3 | $2.4 \times 10^{-10}$ | Overall |
| LINC00886 | −1.7 | $4.2 \times 10^{-2}$ | ER-Neg | Exp-TWAS | −5.0 | $5.9 \times 10^{-7}$ | Overall |
| LRRC25 | 1.9 | $2.8 \times 10^{-2}$ | Overall | Exp-TWAS | 9.5 | $2.8 \times 10^{-21}$ | Overall |
| LRRC37A2 | −1.8 | $3.7 \times 10^{-2}$ | ER-Neg | Exp-TWAS | −5.7 | $1.5 \times 10^{-8}$ | Overall |
| MAN2C1 | −3.3 | $4.7 \times 10^{-4}$ | Overall | Exp-TWAS | −5.9 | $2.8 \times 10^{-9}$ | Overall |
| METTL15P1 | −2.4 | $8.8 \times 10^{-3}$ | ER-Neg | Exp-TWAS | −5.1 | $2.8 \times 10^{-7}$ | Overall |
| MLEC | 2.1 | $1.6 \times 10^{-2}$ | ER-Neg | spTWAS | 5.6 | $2.6 \times 10^{-8}$ | Overall |
| NAGLU | −1.8 | $4.0 \times 10^{-2}$ | Overall | spTWAS | −5.3 | $1.3 \times 10^{-7}$ | Overall |
| NCF1 | 1.8 | $4.0 \times 10^{-2}$ | ER-Pos | APA-WAS | 4.9 | $9.4 \times 10^{-7}$ | Overall |
| PEX14 | 2.3 | $1.2 \times 10^{-2}$ | Overall | Exp-TWAS | 4.9 | $1.2 \times 10^{-6}$ | Overall |
| PLEKHM1 | −1.7 | $4.9 \times 10^{-2}$ | ER-Neg | spTWAS | −5.9 | $4.1 \times 10^{-9}$ | Overall |
| PRSS45 | −2.0 | $2.1 \times 10^{-2}$ | ER-Neg | spTWAS | −5.4 | $6.3 \times 10^{-8}$ | Overall |
| RCCD1 | −2.6 | $4.7 \times 10^{-3}$ | Overall | Exp-TWAS | −10.9 | $1.6 \times 10^{-27}$ | Overall |
| RPS23 | 2.1 | $1.6 \times 10^{-2}$ | Overall | Exp-TWAS | 6.1 | $1.4 \times 10^{-9}$ | Overall |
| SEMA4A | 1.7 | $4.2 \times 10^{-2}$ | TNBC | spTWAS | 6.3 | $4.2 \times 10^{-10}$ | Overall |
| SEPT14P8 | −1.9 | $2.8 \times 10^{-2}$ | ER-Pos | Exp-TWAS | −6.0 | $2.0 \times 10^{-9}$ | ER-Neg |
| SGCE | 1.7 | $4.1 \times 10^{-2}$ | ER-Pos | APA-WAS | 5.7 | $9.3 \times 10^{-9}$ | Overall |
| STXBP4 | 1.9 | $3.1 \times 10^{-2}$ | Overall | spTWAS | 10.0 | $2.2 \times 10^{-23}$ | Overall |
| THBS3 | 2.0 | $2.5 \times 10^{-2}$ | ER-Pos | Exp-TWAS | 5.3 | $1.3 \times 10^{-7}$ | Overall |

*P* values were derived from the *Z* score tests (one-sided).

[a]*ER-Pos* estrogen receptor (ER)-positive, *ER-Neg* ER-negative, *TNBC* triple-negative breast cancer.

[b]*TWAS* transcriptome-wide association study, *Exp-TWAS* gene expression TWAS, *APA-WAS* alternative polyadenylation (APA)-wide association study, *spTWAS* splicing TWAS.

S-PrediXcan tool[43]. The details of the formula used in this method are

$$Z_g \approx \sum_{l \in Model_g} w_{lg} \frac{\hat{\sigma}_l}{\hat{\sigma}_g} \frac{\hat{\beta}_l}{se(\hat{\beta}_l)} \qquad (1)$$

In brief, the Z score was used to estimate the association between predicted transcriptome levels and breast cancer risk. In this formula, $w_{lg}$ is the weight of SNP $l$ for predicting the transcriptome levels of gene $g$. $\hat{\beta}_l$ and $se(\hat{\beta}_l)$ are the association regression coefficient and its standard error for SNP $l$ in GWAS, and $\hat{\sigma}_l$ and $\hat{\sigma}_g$ are the estimated variances of SNP $l$ and the predicted transcriptome levels of gene $g$, respectively. For this study, we estimated the correlations between SNPs included in the prediction models.

**Conditional analyses and permutation tests**

We additionally conducted conditional analyses by adjusting for the nearest GWAS-identified risk signal (the lead SNP, with the strongest association with cancer risk in the locus). For each variant included in the model of genetically predicted levels of gene expression, APA, and exon junction, GCTA-COJO analyses[44] were performed to calculate

the statistical significance with cancer risk after adjusting for the nearest lead variant. We further conducted S-PrediXcan analyses based on the adjusted statistics (i.e., β and SE) to investigate the genetically predicted levels of gene expression, APA, and exon junction in association with cancer risk. The permutation test as described in Gusev et al.'s[32] work was used to assess whether the SNP/gene relationship adds to the SNP/trait associations, with a maximum of 1000 permutations and an initiate permutation *P*-value threshold of 0.05 for each feature.

**Statistical methods**

The primary outcomes of the study are the risk of breast cancer overall and by subtypes. To account for multiple testing, we applied the Bonferroni correction to each association analysis for gene expression (0.05/9982), APA (0.05/3309), and exon junction (0.05/11,426), respectively. To compare the association direction of genes from our study with previous TWAS, we employed a one-tailed *P* value with a significance level of 5% ($P < 0.05$).

**Reporting summary**

Further information on research design is available in the Nature Portfolio Reporting Summary linked to this article.

## Data availability

The genotyping and RNA-Seq data of samples from the Komen Tissue Bank generated in this study have been deposited in the dbGaP database under accession code phs003535.v1.p1 [https://www.ncbi.nlm.nih.gov/projects/gap/cgi-bin/study.cgi?study_id=phs003535.v1.p1]. Summary-level statistics data for the AABCG study are available in the GWAS Catalog under accession number GCST90296719 for overall breast cancer, GCST90296720 for ER-positive breast cancer, GCST90296721 for ER-negative breast cancer, and GCST90296722 for TNBC. GENCODE datasets are available from https://www.gencodegenes.org/human/release_26.html.

## Code availability

The developed pipeline and main source R codes used in this work are available from the GitHub website: https://github.com/pingjie/AABCG_TWAS/[45].

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

## Acknowledgements

The content is solely the responsibility of the authors and does not necessarily represent the official views of the funding agents. The funders had no role in study design, data collection and analysis, decision to publish, or preparation of the manuscript. This research was supported in part by U.S. National Institutes of Health grants R01CA235553, R01CA202981, and Anne Potter Wilson professorship funds. Sample preparation and genotyping assays at Vanderbilt University Medical Center were conducted at the Survey and Biospecimen Shared Resources and Vanderbilt Technologies for Advanced Genomics, which are supported in part by the Vanderbilt-Ingram Cancer Center (P30CA068485). Data analyses were conducted using the Advanced Computing Center for Research and Education (ACCRE) at Vanderbilt University, which is supported in part by the NIH S10 Shared Instrumentation Grant 1S10OD023680. Additional information is provided in the supplemental notes. Research support for participating studies of the AABCG consortium can be found in the Supplement.

## Author contributions

Conception and design of the study: Wei Zheng. Data analyses: Jie Ping. Recruitment of study participants, data and specimen collection: Qiuyin Cai, Christine Ambrosone, Dezheng Huo, Stefan Ambs, Mollie E. Barnard, Yu Chen, Montserrat Garcia-Closas, Jian Gu, Jennifer J. Hu, Esther M. John, Christopher I. Li, Katherine Nathanson, Barbara Nemesure, Olufunmilayo I. Olopade, Tuya Pal, Michael F. Press, Maureen Sanderson, Dale P. Sandler, Toshio Yoshimatsu, Prisca O. Adejumo, Thomas Ahearn, Abenaa M. Brewster, Anselm J.M. Hennis, Timothy Makumbi, Paul Ndom, Katie M. O'Brien, Andrew F. Olshan, Mojisola M. Oluwasanu, Sonya Reid, Song Yao, Ebonee N. Butler, Maosheng Huang, Atara Ntekim, Bingshan Li, Melissa A. Troester, Julie R. Palmer, Christopher A. Haiman, Jirong Long, Wei Zheng. Sample and data preparation or quality control: Jie Ping, Guochong Jia, Xingyi Guo, Ran Tao, Montserrat Garcia-Closas, Jian Gu, Dezheng Huo, Esther M. John, Melissa A. Troester, Song Yao, Thomas Ahearn, Ebonee N. Butler, Maosheng Huang, Qiuyin Cai. Interpretation of findings: Jie Ping, Guochong Jia, Bingshan Li, Jirong Long, Wei Zheng. Drafting or substantively revising the manuscript: Jie Ping, Guochong Jia, Xingyi Guo, Julie R. Palmer, Christopher A. Haiman, Jirong Long, Wei Zheng. Overall supervision of the project: Wei Zheng.

## Competing interests

Olufunmilayo I. Olopade is a co-founder at CancerIQ, serves as a Scientific Advisor at Tempus and is on the Board of 54gene. Atara Ntekim received research support from Roche. The authors declare no competing interests.

## Additional information

Jie Ping ®[1], Guochong Jia ®[1], Qiuyin Cai ®[1], Xingyi Guo ®[1], Ran Tao[2,3], Christine Ambrosone ®[4], Dezheng Huo ®[5], Stefan Ambs ®[6], Mollie E. Barnard[7], Yu Chen[8], Montserrat Garcia-Closas ®[9], Jian Gu ®[10], Jennifer J. Hu ®[11], Esther M. John[12], Christopher I. Li[13], Katherine Nathanson ®[14,15], Barbara Nemesure[16], Olufunmilayo I. Olopade[17], Tuya Pal[18], Michael F. Press ®[19], Maureen Sanderson[20], Dale P. Sandler ®[21], Toshio Yoshimatsu[5], Prisca O. Adejumo[22], Thomas Ahearn ®[9], Abenaa M. Brewster[23], Anselm J. M. Hennis[24,25], Timothy Makumbi[26], Paul Ndom[27], Katie M. O'Brien[21], Andrew F. Olshan[28], Mojisola M. Oluwasanu[29], Sonya Reid[30], Song Yao ®[4], Ebonee N. Butler ®[28], Maosheng Huang[10],

Atara Ntekim ⓘ [31], Bingshan Li ⓘ [32], Melissa A. Troester[28], Julie R. Palmer[7], Christopher A. Haiman ⓘ [33], Jirong Long[1] & Wei Zheng ⓘ [1] ✉

[1]Division of Epidemiology, Department of Medicine, Vanderbilt Epidemiology Center, Vanderbilt-Ingram Cancer Center, Vanderbilt University Medical Center, Nashville, TN, USA. [2]Department of Biostatistics, Vanderbilt University Medical Center, Nashville, TN, USA. [3]Vanderbilt Genetics Institute, Vanderbilt University Medical Center, Nashville, TN, USA. [4]Department of Cancer Prevention and Control, Roswell Park Comprehensive Cancer Center, Elm & Carlton Streets, Buffalo, NY, USA. [5]Department of Public Health Sciences, The University of Chicago, Chicago, IL, USA. [6]Laboratory of Human Carcinogenesis, Center of Cancer Research, National Cancer Institute, National Institutes of Health, Bethesda, MD, USA. [7]Slone Epidemiology Center, Boston University, Boston, MA, USA. [8]Division of Epidemiology, Department of Population Health, NYU Grossman School of Medicine, New York, NY, USA. [9]Division of Cancer Epidemiology and Genetics, National Cancer Institute, Bethesda, MD, USA. [10]Department of Epidemiology, The University of Texas MD Anderson Cancer Center, Houston, TX, USA. [11]Department of Public Health Sciences, University of Miami School of Medicine, Miami, FL, USA. [12]Departments of Epidemiology & Population Health and of Medicine, Stanford University School of Medicine, Stanford, CA, USA. [13]Division of Public Health Sciences, Fred Hutchinson Cancer Center, Seattle, WA, USA. [14]Division of Hematology/Oncology, Department of Medicine, Perelman School of Medicine, University of Pennsylvania, Philadelphia, PA, USA. [15]Basser Center for BRCA, Abramson Cancer Center, Perelman School of Medicine, University of Pennsylvania, Philadelphia, PA, USA. [16]Department of Family, Population and Preventive Medicine, Renaissance School of Medicine, Stony Brook University, Stony Brook, NY, USA. [17]Center for Clinical Cancer Genetics and Global Health, Department of Medicine, The University of Chicago, Chicago, IL, USA. [18]Division of Genetic Medicine, Department of Medicine, Vanderbilt-Ingram Cancer Center, Vanderbilt University Medical Center, Nashville, TN, USA. [19]Department of Pathology, Norris Comprehensive Cancer Center, University of Southern California, Los Angeles, CA, USA. [20]Department of Family and Community Medicine, Meharry Medical College, Nashville, TN, USA. [21]Epidemiology Branch, National Institute of Environmental Health Sciences, National Institutes of Health, Research Triangle Park, NC, USA. [22]Department of Nursing, College of Medicine, University of Ibadan, Ibadan, Nigeria. [23]Department of Clinical Cancer Prevention, The University of Texas MD Anderson Cancer Center, Houston, TX, USA. [24]George Alleyne Chronic Disease Research Centre, University of the West Indies, Bridgetown, Barbados. [25]Department of Family, Population and Preventive Medicine, Stony Brook University, Stony Brook, NY, USA. [26]Department of Surgery, Mulago Hospital, Kampala, Uganda. [27]Yaounde General Hospital, Yaounde, Cameroon. [28]Department of Epidemiology and Lineberger Comprehensive Cancer Center, University of North Carolina at Chapel Hill, Chapel Hill, NC, USA. [29]Department of Health Promotion and Education, College of Medicine, University of Ibadan, Ibadan, Nigeria. [30]Division of Hematology and Oncology, Department of Medicine, Vanderbilt-Ingram Cancer Center, Vanderbilt University Medical Center, Nashville, TN, USA. [31]Department of Radiation Oncology, College of Medicine, University of Ibadan, Ibadan, Nigeria. [32]Department of Molecular Physiology & Biophysics, Vanderbilt Genetics Institute, Vanderbilt University, Nashville, TN, USA. [33]Department of Preventive Medicine, Keck School of Medicine of USC, Los Angeles, CA, USA. ✉e-mail: wei.zheng@vanderbilt.edu

