## [Peer Review File · Nature Communications]

Using genome and transcriptome data from African-ancestry women to identify putative breast cancer susceptibility genesREVIEWER COMMENTS

Reviewer #1 (Remarks to the Author):

Enclosed is a review of Ping et al's manuscript:

"Using genome and transcriptome data from African-ancestry women to identify putative breast cancer susceptibility genes"

In this manuscript, the authors conduct a transcriptome-wide association study of gene expression, exon-junction splicing events and 3' UTR APAs, specifically for woman of African ancestry. The analysis reveals 6 transcriptome-wide significant genes and replicates previous TWAS associations at nominal significance. I commend the authors' commitment to equity in genetic research. The research methods are solid and the results are well communicated. I have some comments point-by-point below.

ABSTRACT

1. This line is a little confusing: "We built genetic models to predict gene expression levels, exon junction splicing events, and 3' UTR alternative polyadenylation for association analyses using genomic data from 18,034 cases and 22,104 controls." It gives the impression that the models were trained in this large dataset. I would clarify or split the sentence.
2. I would include the P-value threshold for replication in the abstract, as well.

INTRODUCTION

1. It could be worth mentioning that previous TWAS of breast cancer mortality and recurrence in CBCS have been conducted by some co-authors (10.1186/s13059-020-1942-6 and 10.1158/0008-5472.CAN-21-1207). This could justify when TWAS for risk in AA women could show differences in germline associations.

RESULTS

1. I would suggest the authors run a permutation test on these 6 Bonferroni-significant genes to assess whether the SNP -> gene relationship adds to the SNP -> trait associations, especially since many of these genes were found at GWAS loci. Gusev 2016's FUSION package adds this as a follow-up test. Briefly, the SNP to gene weights are shuffled randomly and the Z-score is computed for each permutation to generate a null distribution.
2. It may be worth running some conditional analyses to see if the predicted expression of these genes or the splicing event or APA measures can explain away the GWAS loci. This could add some more context to the associations.

Reviewer #2 (Remarks to the Author):

Racial disparity is a significant clinical problem in breast cancer care, with strong biological and clinical evidence suggesting that African American (AA) women face a higher risk of developing triple-negative breast cancer (TNBC). However, the genetic and molecular mechanisms underlying this disparity remain largely unclear, primarily due to the lack of large cohort studies focused on the AA population. Most genome-wide association studies (GWAS) and transcriptome-wide association studies (TWAS) have been conducted on populations of European ancestry (EA). In this context, the authors of this study conducted one of the very first TWAS studies using an AA population and generated a novel resource.

Normal breast tissue samples donated by 150 women of African ancestry were utilized in this study. How was ancestry classified: by self-reported ancestry or based on genetic ancestry analysis? If based on self-reported ancestry, the authors should examine the matched genotyping data to assess genetic ancestry. This analysis was crucial for addressing two key aspects: 1) Consistency between self-

reported ancestry and genetic ancestry: It was essential to determine whether the self-reported ancestry aligned with the genetic ancestry information derived from the genotyping data. 2) Percentage of African ancestry for each individual in the cohort: Since the African American population in the US is an admixed population, it was important to ascertain the proportion of African ancestry in each individual within the cohort.

Through robust TWAS analysis, the authors identified new genes associated with breast cancer in the AA population, including CTD-83 3080P12.3, EN1, LINC01956, and NUP210L. This study sheds new light on the racial disparity observed in breast cancer. Furthermore, a major and significant contribution of this study is the generation of novel and valuable transcriptome profiles from a large cohort of AA individuals. This resource will greatly enhance research on breast cancer disparity in the AA population.

However, although the authors have mentioned that researchers interested in using the data from this study can submit an application to Dr. Wei Zheng, it is not a convenient way to provide access to the resource for the general public. The raw profiling data should be made available through publicly accessible databases, such as NIH dbGaP.

To further investigate the expression and genomic alterations of the four genes (CTD-83 3080P12.3, EN1, LINC01956, and NUP210L) identified in this study, additional analysis should be performed using other breast cancer datasets, such as The Cancer Genome Atlas (TCGA) and the International Cancer Genome Consortium (ICGC). Moreover, it is crucial to explore the expression differences of these genes in AA breast cancer cohorts. Several outstanding datasets based on AA breast cancers are available, such as PMID: 30327465.

Reviewer #3 (Remarks to the Author):

The authors are studying an important problem of identifying genetic contributors of breast cancer, and doing so in an under-researched population of African American (AA) women whose clinical presentation differs from their EA counterparts. To do so, the authors propose a TWAS in AA women to identify breast cancer susceptibility genes. They developed predictive models for gene expression, exon junction splice events and 3' UTR alternative polyadenylation to be used for association upon imputation into genomic data from 18K cases and 22K controls. Using Bonferroni and FDR adjustment for significance, the authors identified 6 breast cancer associated genes, 4 new, and replicated 29 genes reported previously.

GWAS identifies loci, but does not provide information on mechanism or responsible gene while methods such as colocalization analysis, mendelian randomization or TWAS using expression imputation inform this missing information. The authors address the gap that there have been no TWAS studies to identify breast cancer genes in AA women, despite their use in other populations. The contribution of developing novel models for exon junction splicing events and 3' UTR alternative polyadenylation provides important post-transcriptional regulation information in addition to the association with message levels.

Concerns around significance criteria, display of more data including the GWAS data and clarity needed around what is novel and study design, delineated in greater detail below, dampen enthusiasm despite the importance of contributing experimental data and results in this breast cancer population.

1) Often GWAS papers themselves include colocalization or eQTL investigation to try to tackle a putative explanation for the genome-wide significant SNPs. Was this carried out, with a discussion of the loci, in the original AA GWAS (Nat Gen 2023 Under review)? Are the reviewers requesting this if not? And if so, will this current paper provide any novel contribution?

2) Second page of results, first line "Both EN1 and LINC01956 are located at an AA specific risk locus (lead variant rs76664032) that was identified in our recent GWAS...". According to public databases the allele frequency of this SNP in AA is 0 and this SNP is not an eQTL in any GTEx tissue and is not annotated to a gene. It would be very helpful to have the GWAS summary statistics at the loci presented here. For example, visualizing eQTLs and GWAS statistics at the identified GWAS loci provides a visual representation of whether the GWAS loci can be explained by expression variation and for which genes. This would also provide insight into mechanism, as well as the evidence for pleiotropy; that is, are other genes at the loci also demonstrating co-regulation as we know this is a limitation of TWAS. The authors could also provide the same for any SNP level post-transcriptional data and compare and contrast the different breast cancer subtype data. Or show visualizations with the EA breast cancer GWAS and AA GWAS contrasting novel versus common susceptibility loci.

Examples of these tools include eQTLPlot

(<https://biodatamining.biomedcentral.com/articles/10.1186/s13040-021-00267-6>) and LocusFocus

(<https://journals.plos.org/ploscompbiol/article?id=10.1371/journal.pcbi.1008336>) . I encourage the

authors to include visualizations at the novel loci they are proposing here to facilitate discussion on mechanism, pleiotropy, whether they are identifying something novel or a derivative of the GWAS; and then to incorporate these findings into a robust discussion of biology in the discussion section, which is missing.

3) Table 4 likewise investigates genetic variants within 500kb for the 13 genes associated in the present study. Again, a description and visualization of all the SNPs investigated at the 500kb locus rather than just reporting the min p would provide much more information and should be displayed; and the plots could include the GWAS stats by sub-type on the same visualization.

4) Results, 3rd paragraph: "Applying these models to GWAS data, we identified 567 genes associated..." This is out of 9,982 that were tested? And the authors only chose to test genes with $R > 0.1$? This needs to be made clear. And if so, then this is about what would be expected by chance at the 5% significance level. So these 5% results are really not worthy of reporting, and certainly not reporting as "associated."

5) There are association analyses presented by overall breast cancer, by sub-types, and for gene expression, splice events and alternative polyadenylation. But there is no comment/discussion on the primary outcome, choice of significance criteria for each comparison, etc. And this comment is relevant to both new gene identification and replication of previously identified genes in the EA TWAS analysis. The small sample size is understandable, but a comprehensive discussion and clear articulation of significance criteria and justification is needed in the manuscript. And also the FDR adjustment, what comparisons does this refer to?

6) Discussion section: Great that several genes have been replicated from previous Asian or EA TWAS, but I am not sure this provides causal evidence. How about implementing some MR here for causal interpretation?

7) Second last paragraph discussion, it is true that studies using APA-WAS and spTWAS and TWAS can identify genes associated with complex traits, but this is introductory material unless you add some interpretation of your findings as they related to the APA-WAS and spTWAS. What did you learn here from these as a mechanism? Please include in the discussion.

8) Given aberrant expression in tumor tissue, can the authors discuss why expression models from healthy tissue is especially relevant here, and what is being aimed to detect?

9) The last sentence of the discussion states "...and provide new insights into tumor biological differences." But this was not discussed and the results on this were not integrated in the discussion section. Please elaborate and interpret your results within the context of cross-ancestral biological differences.

10) Some discussion on the limitations of TWAS would be warranted as causal interpretations should be taken cautiously. See Wainberg et al Nature Genetics 2019.

- Methods: Significance criteria discussion/justification is missing from the methods section

REVIEWER COMMENTS

Reviewer #1 (Remarks to the Author):

Enclosed is a review of Ping et al's manuscript:

"Using genome and transcriptome data from African-ancestry women to identify putative breast cancer susceptibility genes"

Reviewer: *In this manuscript, the authors conduct a transcriptome-wide association study of gene expression, exon-junction splicing events and 3' UTR APAs, specifically for woman of African ancestry. The analysis reveals 6 transcriptome-wide significant genes and replicates previous TWAS associations at nominal significance. I commend the authors' commitment to equity in genetic research. The research methods are solid and the results are well communicated. I have some comments point-by-point below.*

Response: We appreciate the comments and suggestions and have revised the manuscript accordingly, as elaborated point by point below. **All changes made in the manuscripts are highlighted to facilitate the review of this revised manuscript.**

Reviewer: ABSTRACT

1. This line is a little confusing: "We built genetic models to predict gene expression levels, exon junction splicing events, and 3' UTR alternative polyadenylation for association analyses using genomic data from 18,034 cases and 22,104 controls." It gives the impression that the models were trained in this large dataset. I would clarify or split the sentence.

Response: We thank the reviewer for this helpful suggestion. We have revised this sentence:

Page 3, line 4 - 7:

We built genetic models to predict gene expression levels, exon junction splicing events, and 3' UTR alternative polyadenylation using genomic and transcriptomic data generated in normal breast tissues from 150 AA women and then used these models to perform association analyses using genomic data from 18,034 cases and 22,104 controls.

Reviewer: *2. I would include the P-value threshold for replication in the abstract, as well.*

Response: We appreciate the comment. We have included the P-value threshold:

Page 3, line 11-13:

We also replicated the associations with 29 genes reported in previous TWAS at $P < 0.05$ (one sided), providing further supports for an association of these genes with breast cancer risk.

Reviewer: INTRODUCTION

1. It could be worth mentioning that previous TWAS of breast cancer mortality and recurrence in CBCS have been conducted by some co-authors (10.1186/s13059-020-1942-6 and 10.1158/0008-5472.CAN-21-1207). This could justify when TWAS for risk in AA women could show differences in germline associations.

Response: As suggested, we have revised the introduction and mentioned those previous TWAS of AA breast cancer mortality and recurrence:

Page 4, line 14 - 16:

Two recent TWAS included AA women. However, these two studies were small, evaluated only breast cancer mortality and recurrence and used transcriptomic data from tumor tissues for model building, which could affect the accuracy of model building.^{19,20}

Reviewer: RESULTS

1. I would suggest the authors run a permutation test on these 6 Bonferroni-significant genes to assess whether the SNP -> gene relationship adds to the SNP -> trait associations, especially since many of these genes were found at GWAS loci. Gusev 2016's FUSION package adds this as a follow-up test. Briefly, the SNP to gene weights are shuffled randomly and the Z-score is computed for each permutation to generate a null distribution.

Response: We thank the reviewer for this helpful suggestion. We have performed the permutation test using the method recommended by the reviewer. The total number of permutations performed was set to 1,000. The empirical permutation p-values of all these genes are less than 0.05. The results are shown below and in Table S8.

Breast Cancer Subtype	Gene	Cytoband	Nearest Index SNP	P _{Permutation}	Z _{Conditional}	P _{Conditional}
PrediXcan						
Overall	CTD-3080P12.3	5p15.33	rs2853669	0.001	1.62	0.11
	EN1	2q14.2	rs76664032	< 0.001	-5.03	5.0 × 10 ⁻⁷
ER-Neg	LINC01956	2q14.2	rs76664032	< 0.001	-3.41	6.4 × 10 ⁻⁴
	CTD-3080P12.3	5p15.33	rs2853669	0.001	2.2	0.028
	EN1	2q14.2	rs76664032	< 0.001	-5.07	4.0 × 10 ⁻⁷
TNBC	LINC01956	2q14.2	rs76664032	< 0.001	-3.38	7.1 × 10 ⁻⁴
	MRPL34	19p13.1	rs4609972	0.001	0.23	0.82
APA-WAS						
Overall	TET2	4q24	rs62331150	< 0.001	-1.35	0.18
ER-Neg	TET2	4q24	rs62331150	< 0.001	-0.83	0.4
spTWAS						
Overall	BRD9	5p15.33	rs2853669	< 0.001	-1.7	0.089
	NUP210L	1q21.3	/	< 0.001	/	/
ER-Neg	BRD9	5p15.33	rs2853669	0.002	-1.85	0.065
TNBC	BRD9	5p15.33	rs2853669	< 0.001	-1.14	0.25

Reviewer: 2. It may be worth running some conditional analyses to see if the predicted expression of these genes or the splicing event or APA measures can explain away the GWAS loci. This could add some more context to the associations.

Response: As suggested, we performed conditional analyses by adjusting the nearest index SNPs (at least within ±500Kb region of the gene) using COJO. The NUP210L gene is located far away from any index SNP, and thus, we cannot perform a conditional analysis for this gene. For gene EN1 and LINC01956, we adjusted the SNP rs76664032 identified in our recent AA GWAS. SNP rs76664032 is located at 2q14.2 and is 10Kb from the 3' end of LINC01956 and 18kb from the 3' end of EN1, which is significantly associated with risk of ER negative ($P_{GWAS}=1.44 \times 10^{-9}$) and TNBC ($P_{GWAS}=3.64 \times 10^{-10}$) subtypes in our recent AA GWAS paper. The results of the conditional analysis revealed that the association with EN1 and LINC01956 remain highly significant for the risk of ER-negative (EN1: $P_{TWAS}=1.6 \times 10^{-8}$,

$P_{\text{Conditional}}=5.0\times 10^{-7}$; LINC01956: $P_{\text{TWAS}}=5.3\times 10^{-8}$, $P_{\text{Conditional}}=6.4\times 10^{-4}$) and TNBC subtypes (EN1: $P_{\text{TWAS}}=6.9\times 10^{-9}$, $P_{\text{Conditional}}=4.0\times 10^{-7}$; LINC01956: $P_{\text{TWAS}}=2.5\times 10^{-8}$, $P_{\text{Conditional}}=7.1\times 10^{-4}$) (see table above – also included in Table S8). These results suggest that the association of EN1 and LINC01956 identified in our TWAS cannot be explained entirely by the risk variant identified in GWAS. On the other hand, the association was substantially attenuated for the CTD-3080P12.3 gene ($P_{\text{TWAS}}=1.3\times 10^{-5}$, $P_{\text{Conditional}}=0.028$) or became non-significant for other genes, suggesting that most of the association signal for these genes can be explained by the index variants identified in GWAS. The results are included in the table shown above and in Table S8 and relevant changes are made in the result section (page 6, line 24 – page 7, line 12) and method section (page 12, line 17 – page 13, line 2).

Reviewer #2 (Remarks to the Author):

Racial disparity is a significant clinical problem in breast cancer care, with strong biological and clinical evidence suggesting that African American (AA) women face a higher risk of developing triple-negative breast cancer (TNBC). However, the genetic and molecular mechanisms underlying this disparity remain largely unclear, primarily due to the lack of large cohort studies focused on the AA population. Most genome-wide association studies (GWAS) and transcriptome-wide association studies (TWAS) have been conducted on populations of European ancestry (EA). In this context, the authors of this study conducted one of the very first TWAS studies using an AA population and generated a novel resource.

Normal breast tissue samples donated by 150 women of African ancestry were utilized in this study. How was ancestry classified: by self-reported ancestry or based on genetic ancestry analysis? If based on self-reported ancestry, the authors should examine the matched genotyping data to assess genetic ancestry. This analysis was crucial for addressing two key aspects: 1) Consistency between self-reported ancestry and genetic ancestry: It was essential to determine whether the self-reported ancestry aligned with the genetic ancestry information derived from the genotyping data. 2) Percentage of African ancestry for each individual in the cohort: Since the African American population in the US is an admixed population, it was important to ascertain the proportion of African ancestry in each individual within the cohort.

Response: We appreciate the comments and suggestions. All individuals in this study were determined to have at least 80% of African ancestry using Admixture and we have added this to the method section.

Page 11, line 9 – 10:

All individuals had at least 80% proportion of African ancestry using Admixture, using 1000 Genome samples as reference.

Reviewer: Through robust TWAS analysis, the authors identified new genes associated with breast cancer in the AA population, including CTD-3080P12.3, EN1, LINC01956, and NUP210L. This study sheds new light on the racial disparity observed in breast cancer. Furthermore, a major and significant contribution of this study is the generation of novel and valuable transcriptome profiles from a large cohort of AA individuals. This resource will greatly enhance research on breast cancer disparity in the AA population.

However, although the authors have mentioned that researchers interested in using the data from this study can submit an application to Dr. Wei Zheng, it is not a convenient way to provide access to the resource for the general public. The raw profiling data should be made available through publicly accessible databases, such as NIH dbGaP.

Response: Transcriptomic and genomic data generated from these 150 AA women will be submitted to dbGaP, which should be publicly available early next year. Also, we are in the process of submitting the GWAS data used for our association analyses to the GWAS Catalog, and these data should be publicly

available by the end of this year or early next year. In the meantime, researchers can always request access to the data by submitting an application to Dr. Zheng.

Reviewer: To further investigate the expression and genomic alterations of the four genes (CTD-3080P12.3, EN1, LINC01956, and NUP210L) identified in this study, additional analysis should be performed using other breast cancer datasets, such as The Cancer Genome Atlas (TCGA) and the International Cancer Genome Consortium (ICGC). Moreover, it is crucial to explore the expression differences of these genes in AA breast cancer cohorts. Several outstanding datasets based on AA breast cancers are available, such as PMID: 30327465.

Response: We thank the reviewer for this helpful suggestion and the recommended dataset. We evaluated expression differences of these four genes in the TCGA dataset recommended by the reviewer. The median expression levels of these genes are low. The expression levels for three genes (CTD-3080P12.3, EN1, NUP210L) were higher in adjacent normal tissues than tumor tissues. However, none of the differences was statistically significant, perhaps due to a small sample size. Further research is needed.

Reviewer #3 (Remarks to the Author):

The authors are studying an important problem of identifying genetic contributors of breast cancer, and doing so in an under-researched population of African American (AA) women whose clinical presentation differs from their EA counterparts. To do so, the authors propose a TWAS in AA women to identify breast cancer susceptibility genes. They developed predictive models for gene expression, exon junction splice events and 3' UTR alternative polyadenylation to be used for association upon imputation into genomic data from 18K cases and 22K controls. Using Bonferroni and FDR adjustment for significance, the authors identified 6 breast cancer associated genes, 4 new, and replicated 29 genes reported previously.

GWAS identifies loci, but does not provide information on mechanism or responsible gene while methods such as colocalization analysis, mendelian randomization or TWAS using expression imputation inform this missing information. The authors address the gap that there have been no TWAS studies to identify breast cancer genes in AA women, despite their use in other populations. The contribution of developing novel models for exon junction splicing events and 3' UTR alternative polyadenylation provides important post-transcriptional regulation information in addition to the association with message levels.

Concerns around significance criteria, display of more data including the GWAS data and clarity needed around what is novel and study design, delineated in greater detail below, dampen enthusiasm despite the importance of contributing experimental data and results in this breast cancer population.

1) Often GWAS papers themselves include colocalization or eQTL investigation to try to tackle a putative explanation for the genome-wide significant SNPs. Was this carried out, with a discussion of the loci, in the original AA GWAS (Nat Gen 2023 Under review)? Are the reviewers requesting this if not? And if so, will this current paper provide any novel contribution?

Response: We appreciate the comments and suggestions. Yes, eQTL and colocalization analyses were performed in the original AA GWAS paper. EN1 and LINC01956 (RP11-19E11.1) were suggested by eQTL colocalization analyses. The present study provided additional evidence for these two genes using a different approach. Importantly, the present study discovered two additional genes (CTD-3080P12.3 and NUP210L) that were not previously reported as being associated with the risk of breast cancer. Additionally, conditional analyses show that associations with EN1 and LINC01956 remained highly significant after adjusting for the lead variant in the region, suggesting that the association for these genes cannot be explained by the lead variant.

Reviewer: *2) Second page of results, first line "Both EN1 and LINC01956 are located at an AA specific risk locus (lead variant rs76664032) that was identified in our recent GWAS...". According to public databases the allele frequency of this SNP in AA is 0 and this SNP is not an eQTL in any GTEx tissue and is not annotated to a gene. It would be very helpful to have the GWAS summary statistics at the loci presented here. For example, visualizing eQTLs and GWAS statistics at the identified GWAS loci provides a visual representation of whether the GWAS loci can be explained by expression variation and for which genes. This would also provide insight into mechanism, as well as the evidence for pleiotropy; that is, are other genes at the loci also demonstrating co-regulation as we know this is a limitation of TWAS. The authors could also provide the same for any SNP level post-transcriptional data and compare and contrast the different breast cancer subtype data. Or show visualizations with the EA breast cancer GWAS and AA GWAS contrasting novel versus common susceptibility loci. Examples of these tools include eQTPlot (<https://biodatamining.biomedcentral.com/articles/10.1186/s13040-021-00267-6>) and LocusFocus (<https://journals.plos.org/ploscompbiol/article?id=10.1371/journal.pcbi.1008336>). I encourage the authors to include visualizations at the novel loci they are proposing here to facilitate discussion on mechanism, pleiotropy, whether they are identifying something novel or a derivative of the*

GWAS; and then to incorporate these findings into a robust discussion of biology in the discussion section, which is missing.

Response: We thank the reviewer for this helpful suggestion. This variant has an allele frequency of 0.19 for AA women in the gnomAD database (https://gnomad.broadinstitute.org/variant/2-118823485-A-G?dataset=gnomad_r3) which is consistent with the MAF of this SNP in our GWAS data. The summary statistics data will be made publicly available when the original AA GWAS paper is accepted for publication.

As suggested, the eQTpLot was utilized for the visualization of all putative risk genes identified in this study. We plotted the variants employed in the prediction models, as shown in an example provided below. Figures for all putative risk genes identified in this study can be found in the Supplementary Figures. For many of these genes, variants used in the prediction models showed a highly significant association with breast cancer risk.

Page 7, line 12 – 16:

We utilized eQTpLot³¹ to visualize eQTLs and GWAS statistics for putative risk genes identified in this study. All the variants employed in the prediction models were used in the visualization (Supplementary Figures S1 – S13). For many of these genes, variants used in the prediction models showed a highly significant association with breast cancer risk.

eQTpLot analysis for Breast Cancer (ER-Neg) and EN1 In Breast

Reviewer: 3) Table 4 likewise investigates genetic variants within 500kb for the 13 genes associated in the present study. Again, a description and visualization of all the SNPs investigated at the 500kb locus rather than just reporting the min p would provide much more information and should be displayed; and the plots could include the GWAS stats by sub-type on the same visualization.

Response: In response to the second comment, we have used the eQTLPlot to visualize the GWAS and TWAS stats of the novel loci and these figures are included in the revised manuscript.

Reviewer: 4) Results, 3rd paragraph: “Applying these models to GWAS data, we identified 567 genes associated...” This is out of 9,982 that were tested? And the authors only chose to test genes with $R > 0.1$? This needs to be made clear. And if so, then this is about what would be expected by chance at the 5% significance level. So these 5% results are really not worthy of reporting, and certainly not reporting as “associated.”

Response: Yes, the reviewer is correct. This is out of 9,982 genes with prediction performance $R > 0.1$, and we have made it clear. In the revised manuscript, we used a more stringent p -value (< 0.01) and revised the results as indicated below.

Page 5, line 15 – 21:

Applying these 9,982 models to GWAS data using S-PrediXcan, we identified 121 genes associated with breast cancer risk at $P < 0.01$ (Table S3). We further evaluated the associations between predicted gene expression and risk of breast cancer by subtypes and identified 120, 93, and 85 genes associated with ER-positive, ER-negative, and TNBC subtype at $P < 0.01$, respectively (Table S3).

Page 6, line 4 – 10:

Using these models, we identified 34 APA events associated with breast cancer risk at $P < 0.01$ (Table S4), ... We also found that 122 exon junction events associated with breast cancer risk at $P < 0.01$ (Table S5), ...

Page 6, line 14 – 18:

Analyses by subtypes of breast cancer identified 23, 32, and 27 APA events associated with breast cancer risk at $P < 0.01$ for ER-positive, ER-negative, and TNBC subtype, respectively (Table S4). Through spTWAS, we identified 106, 98, and 106 exon junction events associated with breast cancer risk at $P < 0.01$ for ER-positive, ER-negative, and TNBC subtype, respectively (Table S5).

Reviewer: 5) There are association analyses presented by overall breast cancer, by sub-types, and for gene expression, splice events and alternative polyadenylation. But there is no comment/discussion on the primary outcome, choice of significance criteria for each comparison, etc. And this comment is relevant to both new gene identification and replication of previously identified genes in the EA TWAS analysis. The small sample size is understandable, but a comprehensive discussion and clear articulation of significance criteria and justification is needed in the manuscript. And also the FDR adjustment, what comparisons does this refer to?

Response: We appreciate the comments and now provided a paragraph about the significance criteria and primary outcomes for the study.

Page 14, line 3 - 8:

The primary outcomes of the study are risk of breast cancer overall and by subtypes. We utilized $P < 0.01$ as the initial criterium to assess the statistical significance of our findings. To account for multiple testing, we applied the Bonferroni correction and the false discovery rate (FDR) methods. To compare the association direction of genes from our study with previous TWAS, we employed a one-tailed P value with a significance level of 5% ($P < 0.05$).

Reviewer: 6) Discussion section: Great that several genes have been replicated from previous Asian or EA TWAS, but I am not sure this provides causal evidence. How about implementing some MR here for causal interpretation?

Response: We agree with the reviewer that these findings do not prove causality. However, the consistency of the association between studies conducted in AA and other populations provides supports for possible true associations, particularly given the lead variants in several loci are different between these populations. We have added a few sentences in the discussion section to address this issue.

Reviewer: 7) Second last paragraph discussion, it is true that studies using APA-WAS and spTWAS and TWAS can identify genes associated with complex traits, but this is introductory material unless you add some interpretation of your findings as they related to the APA-WAS and spTWAS. What did you learn here from these as a mechanism? Please include in the discussion.

Response: We appreciate this comment. We have included some discussion about the interpretation related to the APA-WAS and spTWAS.

Page 9, line 12 - 16:

Of note, three genes showing a Bonferroni-corrected significant association were identified in APA-WAS or spTWAS, including one new gene (*NUP210L*) not reported before. As APA-WAS or spTWAS investigate genetic variants related to post-transcriptional regulations, these genetic variants may affect the breast cancer risk of through post-transcriptional and regulatory mechanisms.

Reviewer: 8) Given aberrant expression in tumor tissue, can the authors discuss why expression models from healthy tissue is especially relevant here, and what is being aimed to detect?

Response: The major purpose of our TWAS is to identify breast cancer susceptibility genes, similar to breast cancer GWAS that aim at identifying genetic variants for breast cancer risk. We accomplish this goal in TWAS by comparing predicted gene expressions in normal breast tissues between cases and controls. Gene expressions in tumor tissue are altered by somatic mutations and thus their levels are not less likely to be regulated by germline variants. This will introduce noise and affect the accuracy of model building. Therefore, it is would be less optimal to use data from tumor tissues to build gene prediction models for TWAS to identify breast cancer susceptibility genes. We have included this information in the discussion section.

Page 9, line 20 – 26:

The major purpose of TWAS is to identify breast cancer susceptibility genes, similar to breast cancer GWAS that aim at identifying genetic variants for breast cancer risk. Previous studies used gene expression data to build gene prediction models for TWAS^{9,15,32}. Gene expressions in tumor tissue are altered by somatic mutations and thus their levels are not less likely to be regulated by germline variants, which could introduce noise and affect the accuracy of model building. We used gene

expression data from normal breast tissues to build gene expression prediction models for TWAS, which is a major strength of this study.

Reviewer: 9) *The last sentence of the discussion states "...and provide new insights into tumor biological differences." But this was not discussed and the results on this were not integrated in the discussion section. Please elaborate and interpret your results within the context of cross-ancestral biological differences.*

Response: We have modified the sentence as below:

Page 10, line 4 – 5:

Our findings improve the population diversity in genetic studies for breast cancer and provide new insights into the biology and genetics of breast cancer.

Reviewer: 10) *Some discussion on the limitations of TWAS would be warranted as causal interpretations should be taken cautiously. See Wainberg et al Nature Genetics 2019.*

Response: We appreciate this suggestion. As suggested, we have added some discussion on the limitations of TWAS.

Page 9, line 26 - 29:

However, although TWAS identify genes that are associated with the trait of interest, they do not prove that these genes are causal. Future studies, including functional experiments, are needed to generate additional evidence for causal inference.

Reviewer: - *Methods: Significance criteria discussion/justification is missing from the methods section*

Response: Please see our responses to comment 5 above.

REVIEWER COMMENTS

Reviewer #1 (Remarks to the Author):

The authors have adequately addressed my comments. I recommend publication.

Reviewer #2 (Remarks to the Author):

This manuscript has undergone improvements, with additional information incorporated in this revised version.

The authors mentioned, "All individuals had at least 80% African ancestry using Admixture, referencing the 1000 Genome samples." It would be helpful to know the average proportion of African ancestry in the dataset and how many samples fell below the 80% threshold and were consequently excluded from the study. Considering that the US population is an admixture of ancestries, it's likely that many African American individuals may have less than 80% African ancestry, especially when self-reported race is taken into account. This information should be visually presented with detailed data, perhaps as a supplementary figure.

Regarding the question about validation, the authors presented negative results and cited a small sample size as the reason. Nevertheless, there exist large published sample cohorts, such as the one mentioned in my previous comments PMID: 30327465.

Reviewer #3 (Remarks to the Author):

I thank the authors for addressing the majority of my concerns.

A couple still remain around the significance level used and the rationale by which this was chosen.

1) Previously I noted that the number of significant genes identified out of the 9,982 at the 5% level of 567 was consistent with what would be expected by chance at the 5% level. The authors since changed this to the "more stringent p-value..." 1% level, so now what is reported is consistent with what would be expected by chance at the 1% level but the critique still remains – are these genes at the 1% level worthy of reporting as "associated" when you get the exact number you would expect by chance alone?

2) Previously I requested that a significance criteria discussion/justification be added to the methods section, which the authors did do. However, the justification for the significance levels used are missing. For example, why above did the authors use a 1% level? And the authors in the statistical methods section, results and abstract report using a Bonferroni correction (and the false discovery rate in methods). But how many tests were adjusting for? Why was it not 0.05/ (3,309+11,426+9,982)?

3) There is inconsistency throughout the manuscript of whether a 5% Bonferroni adjusted (not sure how many tests the adjustment was for) criteria was used, a 1% criteria or a FDR adjusted level (Discussion, page 7 line 134). The methods need to be very clear and justified on the significance criteria and why the results reported are "significant."

REVIEWER COMMENTS

Reviewer #1 (Remarks to the Author):

The authors have adequately addressed my comments. I recommend publication.

Reviewer #2 (Remarks to the Author):

This manuscript has undergone improvements, with additional information incorporated in this revised version.

The authors mentioned, "All individuals had at least 80% African ancestry using Admixture, referencing the 1000 Genome samples." It would be helpful to know the average proportion of African ancestry in the dataset and how many samples fell below the 80% threshold and were consequently excluded from the study. Considering that the US population is an admixture of ancestries, it's likely that many African American individuals may have less than 80% African ancestry, especially when self-reported race is taken into account. This information should be visually presented with detailed data, perhaps as a supplementary figure.

Response: We appreciate the comments and suggestions. We obtained samples for this study from the Komen tissue bank. We don't know the number of women with an African-ancestry proportion below 80% in the Komen bank as all samples we obtained have the proportion over 80% as this is a selection criterion. We have included a supplementary table (Table S6) with the ancestry proportions of all the samples utilized for model building from the Komen tissue bank.

Regarding the question about validation, the authors presented negative results and cited a small sample size as the reason. Nevertheless, there exist large published sample cohorts, such as the one mentioned in my previous comments PMID: 30327465.

Response: We applied and obtained the approval from the dbGaP for the WABCS dataset (phs001687, PMID:30327465). However, no RNAseq data are available on either the dbGaP or NCI GDC data portal. We contacted the WABCS PI and were told that these data have not yet been uploaded to the dbGaP due to some delays. We were also told that no transcriptome data from normal/adjacent breast samples are available in the WABCS. With the help of the WABCS team, we were able to include transcriptome data from 133 breast tumor samples in this study. As shown below, the expression levels of these genes in tumor tissues are comparable between WABCS and TCGA after normalization across the two datasets. Additional expression data from normal breast tissues is needed in the future to evaluate difference between normal and tumor tissues.

Reviewer #3 (Remarks to the Author):

I thank the authors for addressing the majority of my concerns.

A couple still remain around the significance level used and the rationale by which this was chosen.

1) Previously I noted that the number of significant genes identified out of the 9,982 at the 5% level of 567 was consistent with what would be expected by chance at the 5% level. The authors since changed this to the “more stringent p-value...” 1% level, so now what is reported is consistent with what would be expected by chance at the 1% level but the critique still remains – are these genes at the 1% level worthy of reporting as “associated” when you get the exact number you would expect by chance alone?

Response: We appreciate the comments and suggestions. We have adjusted this statement and reported only the genes with Bonferroni-corrected significant level of 0.05.

Page 4 line 15 – 22:

Applying these 9,982 models to GWAS data using S-PrediXcan, one lncRNA gene CTD-3080P12.3 (also known as TERLR1) was associated with overall breast cancer risk at the Bonferroni-corrected significance of $P < 5.0 \times 10^{-6}$ (0.05/9,982). We further evaluated the associations between predicted gene expression and risk of breast cancer by subtypes and identified two genes EN1 and LINC01956

(for ER-negative and TNBC) showing an association at the Bonferroni-corrected significance level (Table 2). These three genes (CTD-3080P12.3, EN1 and LINC01956) have not been previously identified in TWAS for breast cancer risk.

Page 4 line 26 – Page 6 line 4:

Using these models, we identified one APA event in gene *TET2* that showed an association with overall breast cancer risk at the Bonferroni-corrected significance of $P < 1.5 \times 10^{-5}$ (0.05/3,309, Table 2). We also found that two exon junction events in genes *BRD9* and *NUP210L* showed association with overall breast cancer risk at the Bonferroni-corrected significance of $P < 4.3 \times 10^{-6}$ (0.05/11,426, Table 2).

Page 5 line 6 – 10:

Analyses by subtypes of breast cancer identified *TET2* from APA-WAS in association with ER-negative breast cancer risk at the Bonferroni-corrected significant level and *BRD9* from spTWAS in association for ER-negative and TNBC subtypes at the Bonferroni-corrected significant level (Table 2).

2) Previously I requested that a significance criteria discussion/justification be added to the methods section, which the authors did do. However, the justification for the significance levels used are missing. For example, why above did the authors use a 1% level? And the authors in the statistical methods section, results and abstract report using a Bonferroni correction (and the false discovery rate in methods). But how many tests were adjusting for? Why was it not 0.05/ (3,309+11,426+9,982)?

Response: We appreciate the comment. We applied the Bonferroni correction to each association analysis using expression (0.05/9982), APA (0.05/3309), and exon junction events (0.05/11426). We have revised the method part for Statistical Methods:

Page 12 line 4 – 6:

To account for multiple testing, we applied the Bonferroni correction to each association analysis for gene expression (0.05/9,982), APA (0.05/3,309), and exon junction events (0.05/11,426), respectively.

3) There is inconsistency throughout the manuscript of whether a 5% Bonferroni adjusted (not sure how many tests the adjustment was for) criteria was used, a 1% criteria or a FDR adjusted level (Discussion, page 7 line 134). The methods need to be very clear and justified on the significance criteria and why the results reported are “significant.”

Response: We appreciate the comments. We have adjusted this statement and reported only the genes with Bonferroni-corrected significant level of 0.05.

Page 6 line 20 – 22:

We identified six genes associated with risk of breast cancer, overall or by subtypes, at the Bonferroni-corrected significant level of 0.05, including four genes not yet reported previously.

REVIEWERS' COMMENTS

Reviewer #2 (Remarks to the Author):

My first question has been carefully addressed. Although the second question was not addressed, I think the manuscript has been improved, and I would like to suggest to accept for publication.

Reviewer #3 (Remarks to the Author):

The authors have now adequately addressed my comments.